# Ischemia and Reperfusion Injury in Kidney Transplantation: Relevant Mechanisms in Injury and Repair

**DOI:** 10.3390/jcm9010253

**Published:** 2020-01-17

**Authors:** Gertrude J. Nieuwenhuijs-Moeke, Søren E. Pischke, Stefan P. Berger, Jan Stephan F. Sanders, Robert A. Pol, Michel M. R. F. Struys, Rutger J. Ploeg, Henri G. D. Leuvenink

**Affiliations:** 1Department of Anesthesiology, University of Groningen, University Medical Centre Groningen, Hanzeplein 1, 9713 GZ Groningen, The Netherlands; m.m.r.f.struys@umcg.nl; 2Clinic for Emergencies and Critical Care, Department of Anesthesiology, Department of Immunology, Oslo University Hospital, 4950 Nydalen, 0424 Oslo, Norway; s.e.pischke@medisin.uio.no; 3Department of Nephrology, University of Groningen, University Medical Centre Groningen, Hanzeplein 1, 9713 GZ Groningen, The Netherlands; s.p.berger@umcg.nl (S.P.B.); j.sanders@umcg.nl (J.S.F.S.); 4Department of Surgery, University of Groningen, University Medical Centre Groningen, Hanzeplein 1, 9713 GZ Groningen, The Netherlands; r.pol@umcg.nl (R.A.P.); rutger.ploeg@nds.ox.ac.uk (R.J.P.); h.g.d.leuvenink@umcg.nl (H.G.D.L.); 5Department of Basic and Applied Medical Sciences, Ghent University, Corneel Heymanslaan 10, 9000 Ghent, Belgium; 6Nuffield Department of Surgical Sciences, University of Oxford, Headington, Oxford OX3 9DU, UK

**Keywords:** ischemia reperfusion injury, kidney transplantation, delayed graft function, innate immune system, adaptive immune system, apoptosis, necrosis, hypoxic inducible factor, endothelial dysfunction

## Abstract

Ischemia and reperfusion injury (IRI) is a complex pathophysiological phenomenon, inevitable in kidney transplantation and one of the most important mechanisms for non- or delayed function immediately after transplantation. Long term, it is associated with acute rejection and chronic graft dysfunction due to interstitial fibrosis and tubular atrophy. Recently, more insight has been gained in the underlying molecular pathways and signalling cascades involved, which opens the door to new therapeutic opportunities aiming to reduce IRI and improve graft survival. This review systemically discusses the specific molecular pathways involved in the pathophysiology of IRI and highlights new therapeutic strategies targeting these pathways.

## 1. Introduction

To date, 10% of the worldwide population suffers from chronic kidney disease (CKD). The prevalence of the disease will most likely grow over the next decade due to the increase in the elderly population and the growing incidence of diabetes and hypertension. In 2015, CKD was ranked 12th in the global list of causes of death [1]. The population of patients needing renal replacement therapy (RRT) worldwide was estimated to be approximately 4.902 million (95% CI 4.438–5.431 million) in a conservative model and 9.701 million (95% CI 8.544–11.021 million) in a high estimate model, illustrating the magnitude of the disease burden of end stage renal disease (ESRD) [2]. 

For patients with ESRD, transplantation is still the optimal treatment. Long-term survival with kidney transplantation is dramatically better than dialysis and transplantation provides a sustainably higher quality of life. Unfortunately, there is a worldwide shortage of suitable donor organs for (kidney) transplantation. The number of renal transplantations performed worldwide in 2018 was 75.664 [3]. Due to the persistent shortage of donor kidneys, many transplant centres have established large living donor programmes and transplant teams are also now accepting increasing numbers of older and higher risk organs, retrieved from deceased donors. The use of these extended criteria donors (ECD) has affected outcomes after transplantation due to an often-suboptimal quality of the donor organ [4,5]. As we will face more complex donors in the future with a reduced viability such as unstable donation after brain death (DBD) donors, donation after circulatory death (DCD) donors, and ECD, the challenge in transplantation is to be able to use these donor sources, however, without compromising successful immediate function and long-term graft survival after transplantation. It is therefore imperative that the condition of every graft-to-be is optimised prior to or at the time of transplantation and that additional injury is minimized in order to achieve the best possible post-transplant function and avoid primary non function (PNF), delayed graft function (DGF), and rejection with chronic graft failure.

Ischemia and reperfusion injury (IRI) is inevitable in (kidney) transplantation and one of the most important mechanisms for non- or delayed function immediately after transplantation [6,7,8]. It is accompanied by a proinflammatory response and is associated with acute rejection due to an increased immunogenicity favouring T-cell mediated rejection as well as anti-body mediated rejection (ABMR) [9,10]. In addition, it may result in progressive interstitial fibrosis and is associated with chronic graft dysfunction due to interstitial fibrosis and tubular atrophy (IFTA) [11]. In the past decade more insight has been gained in the complex molecular pathophysiology of IRI. This may open a door to new therapeutic targets aiming to reduce IRI. The aim of this review is to systematically highlight these molecular mechanisms and to discuss potential therapeutic strategies specifically targeting these molecular pathways.

## 2. Ischemia and Reperfusion Injury

IRI consists of a complex pathophysiology involving activation of cell death programs, endothelial dysfunction, transcriptional reprogramming and activation of the innate and adaptive immune system [8]. Numerous pathways and signalling cascades are implicated (Figure 1) and it is while worthy to dissect the distinct effects of ischemia and reperfusion (I/R).

### 2.1. Ischemia 

Due to a decrease in oxygen supply, cells will switch from an aerobic to an anaerobic metabolism, which results in a decrease in adenosine triphosphate (ATP) production and intracellular acidosis due to the formation of lactate. This causes destabilisation of lysosomal membranes with leakage of lysosomal enzymes, breakdown of the cytoskeleton and inhibition of membrane-bound Na^+^/K^+^ ATPase activity [12,13,14]. This last process gives rise to an intracellular accumulation of Na^+^ ions and water with as a consequence cellular oedema. Due to declined Ca^2+^ excretion, there is an intracellular Ca^2+^ accumulation, which causes activation of Ca^2+^ dependant proteases like calpains. Due to the acidosis, these calpains stay inactive during the ischemic period but may damage the cell after normalisation of the pH during reperfusion. The remaining ATP is broken down to hypoxanthine, which will accumulate in the cell, since further metabolism into xanthine requires oxygen [15]. In the mitochondria, the Ca^2+^ overload is responsible for generation of reactive oxygen species (ROS) [8]. This will lead to opening of the mitochondrial permeability transition pores (mPTP) after reperfusion. During the ischemic period, only small amounts of ROS are produced compared to the entire I/R due to the reduction of cytochromes, nitric oxide synthases, xanthine oxidase and reduced nicotinamide adenine dinucleotide phosphate (NADPH) oxidase activation [16,17,18,19].

### 2.2. Reperfusion

During reperfusion, oxygen levels increase, and the pH normalises which is harmful for the previously ischemic cells. The intracellular Ca^2+^ level further increases, which activates the calpains causing injury to the cell structure and cell death [8]. Restoration of normoxemia leads to the production of large amounts of ROS, together with a reduction in the antioxidant capacity [20]. This burst of ROS production was thought to be due to a generalised dysregulation of the electron transport chain with electrons leaking out at non-specific sites [21]. Recently, however, Chouchani et al. [22] showed that this superoxide production is generated by reverse action of complex I of the electron transport chain driven by a pool of succinate, a metabolite of the citric acid cycle, accumulated during ischemia. This massive amount of mitochondrially produced ROS is responsible for the activation of various injurious pathways through carbonylation of proteins or lipid peroxidation. This may contribute to injury of the cell membranes, the cytoskeleton and DNA and may lead to a disruption of ATP generation and induction of mPTP [20]. Additionally, the combination of ROS, dysfunctioning of the mitochondrial machinery and increase in mitochondrial Ca^2+^ load causes opening of the mPTP and release of substances like cytochrome C, succinate and mitochondrial DNA (mtDNA), which are able to induce cell death through apoptosis and necrosis and may act as danger/damage associated molecular patterns (DAMPs) entailing activation of the innate and subsequently the adaptive immune system [23,24,25,26]. 

Recent insights in the pathophysiological mitochondrial mechanisms and general understanding of the pivotal role of the mitochondria in IRI has led to various strategies targeting mitochondria with the aim to reduce IRI including limiting oxidative stress and mitochondrial ROS generation [20]. Both lipophilic cations and mitochondrial targeted proteins have been developed to deliver antioxidants to the mitochondria [27]. Triphenylphosphonium (TTP), a lipophilic cation, is rapidly taken up by mitochondria where it releases covalently bonded bioactive compounds. MitoQ, with its bioactive compound ubiqinone, is the most investigated of these molecules. In the mitochondria ubiqinone is reduced to ubiquinol, a powerful ROS scavenger. Administration of MitoQ in renal I/R models resulted in reduced markers of oxidative stress, reduced renal injury and improved function [28,29,30]. Regarding the mitochondrial targeted proteins, the Szeto-Schiller (SS) proteins are the best known. Exact mechanism of action is poorly understood but a possible explanation of action is through interaction with cardiolipin, an important component of the inner mitochondrial membrane. SS peptides have shown to reduce renal IRI in rodents [31], and its lead compound SS-31 (Elamipretide, Stealth BioTherapeutics-Alexion Pharmaceuticals) is currently being investigated in humans for its efficacy in reducing IRI post-angioplasty for renal artery stenosis. A pilot study administration of SS-31 before and during percutaneous transluminal renal angioplasty and stenting has shown to attenuate post-procedural hypoxia, increased renal blood flow and improved kidney function [32].

Another strategy to reduce ROS generation is reduction of succinate formation by inhibition of succinate dehydrogenase, preventing the accumulation of succinate, a driving force of reverse action of complex I. This has been shown to be effective in various in vivo models of IRI including the heart but has yet been unexplored in renal IRI [22,33].

## 3. Pathophysiological Consequences of IRI

### 3.1. Cell Death: Necrosis, Apoptosis, Regulated Necrosis and Autophagy

#### 3.1.1. Necrosis

I/R leads to the activation of cell death programs. Of these programs, necrosis is the most uncontrolled form. It is due to swelling of the cell and subsequent rupture of the cellular membrane [34]. This will lead to an uncontrolled release of cellular fragments into the extracellular space. These fragments act as DAMPs and are able to activate the innate and adaptive immune system, entailing infiltration of inflammatory cells into the tissue and release of different cytokines. 

#### 3.1.2. Apoptosis

In contrast to the uncontrolled process of necrosis, apoptosis is a highly regulated and controlled process in which activation of the caspase signalling cascade results in a self-limiting programmed cell death. These caspases, a family of proteases, are essential in this process. There are two types of caspases: initiator caspases (2,8,9,10) and effector caspases (3,6,7) [35,36]. The initiator caspases are activated by binding to a specific activator protein complex (death-inducing signalling complex (DISC), apoptosome) [37]. The formed complexes then activate the effector caspases through proteolytic cleavage upon which these proteolytically degenerate various intracellular proteins. Apoptosis gives rise to apoptotic bodies, containing these intracellular protein fragments, via the process of membrane blebbing. The apoptotic bodies will undergo phagocytosis before they can spill their content into the extracellular space and therefore will generate a less immune stimulating impulse compared to necrosis. Apoptosis can be initiated through the intrinsic pathway (mitochondrial dependent pathway) in which the initiating signal comes from within the cell (e.g., damaged DNA, hypoxia, metabolic stress) or the extrinsic pathway (cell death receptor pathway) due to signals from out of the cell (tumor necrosis factor-α (TNF-α), first apoptosis signal (Fas)-ligand, FasL) (Figure 2) [37]. 

A protein family playing an important role in the regulation of apoptosis is the B-cell lymphoma 2 (BcL-2) family [38]. Members of this family can act as protectors (BcL-2, BcL-xL) inhibiting apoptosis, sensors (BH3 only proteins, Bad, Bim, Bid) inhibiting the protectors, or effectors (Bax, Bad) initiating apoptosis by enhancing the permeability of the mitochondrial membrane [39]. In case of intrinsic signalling, intracellular signals of cell stress will lead to an increase in the BH3 only proteins resulting in inhibition of the protectors and activation of the effectors. These effectors increase the permeability of the mitochondrial membrane resulting in leakage of pro-apoptotic proteins upon which a caspase activator complex, the apoptosome, is formed in the intracellular space [40,41,42,43]. The apoptosome cleaves procaspase-9 to its active form of caspase-9, which in turn is able to activate the effector caspase-3. In case of the extrinsic signalling, binding of TNF-α (TNF path) or the FasL, expressed on cytotoxic T lymphocytes, (Fas path) to receptors of the TNF receptor (TNFR) family will lead to the formation of a complex called the death-inducing signalling complex (DISC) [44,45,46]. The DISC, amongst others, consisting of a death effector domain and three procaspase-8 or -10 molecules, cleaves and activates the procaspases [47]. Activation of the initiator caspase-8 by both paths directly activates other members of the caspase signalling cascade such as the effector caspase-3 but also can lead to an increase in BH3-only proteins (Bim, Bid) and trigger the intrinsic pathway (Figure 2) [48].

#### 3.1.3. Regulated Necrosis

Recently, new pathways of a more regulated form of necrosis have been described. These processes show features of apoptosis as well as necrosis. One of the best-known pathways of regulated necrosis is via TNFR-1 and is called necroptosis [46]. In the absence of active caspase-8, phosphorylation of receptor-interacting protein kinase 1 (RIPK1) and RIPK3 in complex IIb leads to formation of a complex called the necrosome. The necrosome recruits Mixed Kinase Domain-Like protein (MLKL), which is then phosphorylated by RIPK3 [46]. MLKL activates the necrosis phenotype by entering the bilipid membranes of organelles and the cellular membrane. This causes formation of pores in these membranes and leads to release of cellular contents, functioning as DAMPs, into the extracellular space [49]. As in necrosis the DAMPs are able to activate both the innate and adaptive immune system promoting proinflammatory responses that activate rejection pathways [50,51]. A recent study in a kidney transplant mouse model showed that RIPK3-deficient kidneys had better function and longer rejection-free survival [52]. Therefore RIPK3-inhibiting drugs might be of interest in the reduction of IRI in organ transplantation. Next to TNFR-1, other death receptors and toll like receptors (TLR) have also shown to be able to induce necroptosis [46]. Other forms of regulated necrosis include mitochondrial permeability transition (MPT)-associated death (involving opening of mPTP leading to necrosis instead of apoptosis), ferroptosis (involving iron and gluthation metabolism), parthanatos (also known as PARP-1 (Poly(ADP-ribose) polymerase-1) dependent cell death, involving the accumulation of PAR (poly(ADP-ribose)) and the nuclear translocation of apoptosis-inducing factor (AIF) from mitochondria) and pyroptosis (involving caspase-1 and -11 in mice and caspase-4 and -5 in humans) [53]. The role of pyroptosis in IRI in the kidney, however, is unclear. 

#### 3.1.4. Autophagy

Cells can preserve their metabolic function and escape cellular death. This is due to autophagy of damaged cell parts. There are several pathways of autophagy, namely, macro-autophagy, micro-autophagy and chaperone-mediated autophagy—the last two are beyond the scope of this review. Macro-autophagy (hereafter called autophagy) involves formation of autophagosomes containing damaged cell parts or unused proteins. These double membrane autophagosomes travel through the cytoplasm to fuse with lysosomes (autolysosome) leading to degradation of the damaged cell parts. This process is continuously active at low basal levels, preserving cellular homeostasis, but stimulated upon stress through various signals like nutrient deprivation, ROS formation, hypoxia, free amino acids, etc. [54,55,56]. Cellular building blocks obtained from recycling of damaged cell parts by autophagy may serve as anti-stress responses and energy source promoting cell survival.

The first step in autophagy, the initiation, is regulated by two kinases: mammalian target of rapamycin complex 1 (mTOR, mTORC1) and adenosine monophosphate-activated protein kinase (AMPK) [54,57,58]. Together, they regulate the activity of the Unc-51 like autophagy activating kinase 1/2 (ULK1/2) complex [59,60]. Activation of mTOR leads to the phosphorylation of this complex and inhibition of autophagy (for instance, through the phosphatidylinositol 3-kinase (PI3K)/Protein kinase B (AKT) or the mitogen-activated protein kinase (MAPK)/extracellular signal–regulated kinase (Erk) 1/2 signalling pathway). On the other hand, activation of AMPK, upon intracellular AMP increase, activates autophagy [61]. This occurs by inhibition of the mTORC1 through dissociation of mTORC1 from ULK1/2 (indirect) or in a direct way by phosphorylation of ULK1/2 forming the ULK1/2-complex [62,63]. Next to the ULK1/2 complex, inducible beclin-1 complex (or class III PI3K complex) is involved in initiation of autophagy. This complex is activated by the ULK-1/2 complex and inhibited by Bcl-2 and Bcl-XL. The ULK1/2 and class III PI3K complexes join to form the phagopore and eventually the autophagosme which will fuse with a lysosome [64,65,66,67,68,69]. The content of this formed autolysosome is degenerated, and the components are released to be reused to synthesise new proteins or to function as an energy source for the cell (Figure 3) [70]. 

In renal IRI, autophagy is considered a doubled-edged sword. Upon I/R, it is mostly upregulated, but both protective and harmful effects are observed, proposing a dual role for autophagy in renal IRI [71,72]. Decuypere et al. [71] hypothesize that autophagy can switch roles depending on the severity of the ischemic injury. The exact mechanism behind this switch is unclear but may depend on the survival vs death properties of beclin1 and its interaction with the Bcl-2 family proteins [71,73]. Autophagy can be considered a protective mechanism in (oxidative) stress injured cells through restoring cellular homeostasis. Kidneys from older donors are at increased risk of DGF. The age-dependent decline in autophagy activity and age-dependant autophagic dysfunction may be one of the underlying mechanisms of this phenomenon [74]. Extensive oxidative stress (amount or duration), however, may have detrimental effects which eventually could trigger the switch to aggravation of the injury through autophagy dependant cell death. Excessive or prolonged ROS exposure may lead to the oxidative modification of macromolecules making them only partially degradable by the autolysosome [75]. Furthermore, an energy dependent process of autophagy could deprive the cell of necessary energy. In this light, excessive autophagy seen after prolonged cold ischemia time in particular in DCD donors seems to be one of the underlying mechanisms behind augmentation of reperfusion injury seen in these circumstances, thereby increasing the risk of DGF [71,76]. Based on this dual role of autophagy in renal IRI and transplantation the goal would be to restrict autophagy levels within a protective window. Upon severe ischemia (prolonged cold ischemia time (CIT)) autophagy inhibitors most likely outweigh the activators [71]. Continuing efforts have to be made to elucidate the mechanism of autophagic transition from protective to harmful function.

The different cell death programs described above are induced in response to common stimuli. Several proteins in the autophagy and apoptosis pathway are shared resulting in an intimate crosstalk between apoptosis and autophagy. Regulation of these proteins determines cellular fate to cell survival or cell death. Caspase-mediated degradation of several autophagy regulation proteins limits autophagosome formation and therefore autophagy [77,78,79]. Apoptosis inhibitors Bcl-2 and Bcl-XL also inhibit autophagy by binding to Beclin-1 limiting its availability to form the classIII PI3K complex [80,81]. Inhibition of cisplatin induced autophagy enhanced caspase-3 activation and apoptosis in renal proximal tubular cells [82,83]. On the other hand, overexpression of ATG5 and beclin-1 prevented cisplatinum induced caspase activation and apoptosis [84]. Additionally, there is evidence that autophagy induction regulates necroptosis. Inhibition of autophagy has shown to prevent necroptosis and vice versa inhibition of necroptosis is able to supress autophagy [85,86].

#### 3.1.5. Targeting Cell Death Programs

Targeting pathways of cell death programs to reduce IRI seems very attractive, since it directly preserves cellular function. Secondly, dead cells releasing DAMPs elicit a strong immune response not only in the organ exposed to I/R but also in other organs of the individual, so called remote organ injury. Therefore, interfering with this process might be immunosuppressive and organ protective. The relative contribution of each of the cell death programs to IRI and outcome in transplantation, however, has to be elucidated. 

Nydam et al. [87] showed in a syngeneic mouse transplant model that administration of the pan-caspase inhibitor Q-VD-OPh during graft retrieval and cold preservation resulted in decreased caspase-3 expression and activity, reduced apoptosis in renal tubular cells and improved renal function post-transplantation. The pro-apoptotic gene p53 is activated upon hypoxia, oxidative stress and DNA damage and is able to induce cell cycle arrest, which enables DNA-repair proteins to repair the sustained injury. However, in case of severe DNA damage it induces apoptosis by initiating the intracellular pathway. 

Inhibition of P53 in proximal tubular cells has been shown to decrease apoptotic cell death and provide protection against IRI [88,89]. QPI-1002 is a synthetic small interfering ribonucleic acid (siRNA) designed to reversibly and temporarily inhibit p53. In pre-clinical models it has been shown that QPI undergoes rapid glomerular filtration and uptake by proximal tubular epithelial cells [89]. Administration of QPI-1002 has shown to be safe in humans. Two phase I dose escalating safety and pharmacokinetics studies in patients undergoing major cardiovascular surgery (NCT00554359, NCT00683553) has been executed without dose-limiting toxicities or safety issues. A phase I/II study has been executed to evaluate QPI-1002 for the prevention of DGF in recipients of kidneys from deceased donors (NCT00802347) in which treatment with QPI-1002 resulted in lower incidence and severity of DGF [90]. Recently, a phase 3 randomized, double-blind, placebo-controlled study in recipients (*n* = 594) of (older) DBD donor kidneys (>45 years) has been completed (NCT02610296, ReGIFT-study). Results have not been reported yet. 

Various pharmacological substances like necrostatins (RIPK1 inhibitors, necroptosis), ferrostatins (ferroptosis), sanglifehrin A (MPT-associated death) and olaparib (parthanatos) and many others have been developed to target specific key molecules of the different programs of regulated necrosis and are currently tested in various animal and disease models (Figure 4) [91,92]. The question remains how safe it will be to inhibit non-apoptocic cell death pathways in patients, since these pathways also function as a backup system when apoptosis fails or is inhibited for instance, by caspase inhibitor expressing viruses. Of these molecules, RIPK1 inhibitors have now entered clinical trials and their safety is being tested in healthy volunteers [93,94].

### 3.2. Endothelial Dysfunction

At a vascular level, I/R leads to swelling of the endothelial cells (ECs), loss of the glycocalyx and degradation of the cytoskeleton. As a consequence, intercellular contact of endothelial cells is lost, increasing vascular permeability and fluid loss to the interstitial space [95]. Furthermore, the endothelium will produce vasoactive substances like platelet-derived growth factor (PDGF) and Endothelin-1 (ET-1), causing vasoconstriction [96]. This vasoconstriction can be enhanced by a reduced nitric oxide (NO) production during reperfusion due to decreased endothelial nitric oxide synthase (eNOS) expression and increased sensitivity of the arterioles for vasoactive substances like angiotensin II, thromboxane A2 and prostaglandin H2 [97,98,99]. Eventually this can lead to the so called no reflow phenomenon characterized by the absence of adequate perfusion on microcirculatory level despite reperfusion.

The regenerative capacity of ECs in peritubular capillaries is limited and injury to the microcirculation may lead to permanent peritubular capillary rarefaction [100,101]. Chronic hypoxia in these regions may induce transcription of fibrogenic genes like transforming growth factor-β (TGF-β) and connective tissue growth factor (CTGF) together with an accumulation of α-smooth muscle actin (α-SMA) [101]. In the end, this may lead to development of IFTA, a process which has mainly been attributed to resident fibroblasts. More recently, however, the role of endothelial-to-mesenchymal transition (EndMT) in this process has been described [102,103]. During EndMT, ECs lose their endothelial phenotype (such as expression of specific endothelial markers like Von Willebrand factor (VWF)) and acquire the phenotype of multipotent mesenchymal cells (MSC). These cells show an increased expression of α-SMA, neuronal (N)-cadherin, vimentin and fibroblast-specific protein-1 and exhibit enhanced migratory potential and increased extracellular matrix production [104,105,106]. In a porcine I/R model Curci et al. [102] showed that 20%–30% of the total α-SMA+ cells emerging after IRI were also CD31+ suggesting a different origin compared to resident activated fibroblasts. Man et al. [107] showed that in kidney transplant recipients experiencing IFTA and allograft dysfunction, progression of EndMT plays an important role. EndMT is controlled by complex signalling pathways and networks. In their porcine I/R model, Curci et al. [102] showed a critical role of complement in this process. Kidneys of pigs treated with recombinant C1 inhibitor (C1-INH) showed preserved EC density, significant reduction of α-SMA expression and limited collagen deposition 24 h after I/R compared to untreated pigs. The ECs in the treated pigs showed preserved physiological conformation and position tight to the basal layer of the vessels. The number of transitioning ECs was significantly lower in the treated animals. In an additional in vitro experiment activating ECs with the anaphylatoxin C3a, they showed that C3a induced down regulation of the expression of VWF whilst upregulating α-SMA, by activating the Akt pathway. Activation of the ECs with C5a showed a similar response [102]. Targeting signalling pathways in EndMT in kidney transplantation could be of interest to reduce IFTA and enhance long-term graft survival. More insight however has to be gained to the exact role of EndMT in renal transplantation and what suitable targets to aim for. Furthermore, since EndMT gives rise to multipotent MSC this placidity could be of interest to push these MSCs in the direction of regeneration rather than fibrosis. 

An important feature of IRI is the chemotaxis of leukocytes, endothelial adhesion and transmigration of these cells into the interstitial compartment [108]. This process is initiated by increased expression of P-selectin on the endothelial cells and interaction of P-selectin with P-selectin glycoprotein 1 (PSGL-) expressed on the leukocytes. This interaction results in rolling of the leukocytes on the endothelium. Subsequently, firm adherence of the leucocytes to the endothelium is achieved by the interaction of the β2-integrins lymphocyte function-associated antigen 1(LFA-1) and macrophage-1 antigen (MAC-1 or complement receptor 3, CR3) on the leukocyte and the intracellular adhesion molecule 1 (ICAM-1) on the endothelial cells. Platelet endothelial cell adhesion molecule 1 (PECAM-1) thereafter facilitates transmigration into the interstitial space. Once activated, these leukocytes will release several toxic substances like ROS, proteases, elastases and different cytokines in the interstitial compartment which will result in further injury like increased vascular permeability, oedema, thrombosis and parenchymal cell death (Figure 5) [109]. 

### 3.3. Innate and Adaptive Immune Response

IRI is accompanied by sterile inflammation in which the innate as well as the adaptive immune system are involved.

#### 3.3.1. Innate Immune Response

The innate, or non-specific, immune system is evolutionary the oldest part of the immune system. It acts on infection or injury with a fast, short-lasting and non-specific response in which different cells and systems are involved. 

##### Toll-Like Receptor Signalling

In the innate immune response, the toll-like receptors (TLRs) play an important role [110]. TLRs are transmembrane proteins and members of the interleukin-1 receptor (IL-IR) superfamily. They function as pattern recognition receptors (PRR) and are present on the cellular membrane and in the cytosol of cells like leukocytes, endothelial cells and tubular cells [111]. The human TLR family contains 10 members, TLR1–TLR10 [112]—of which, TLR2 and TLR4 have shown to be upregulated in tubular epithelial cells upon ischemia [113,114,115,116,117]. Both are attributed an equal importance in initiating apoptosis in a genetic knock-out renal I/R mouse model [115]. TLR activation leads to the downstream recruitment of various adapter molecules (TNF receptor-associated factor 6 (TRAF6), Myeloid differentiation primary-response protein 88 (MyD88), toll-interleukin 1 receptor (TIR) domain containing adaptor protein (TIRAP), TIR-domain-containing adapter-inducing interferon-β (TRIF), TRIF-related adaptor molecule (TRAM)) activating different kinases (IL-1 receptor-associated kinase (IRAK)-1 (IRAK-1), IRAK-4, inhibitor of nuclear factor-κB kinase (IKK), TANK-binding Kinase-1 (TBK1)), leading to activation of transcription factors (nuclear factor kappa-light-chain-enhancer of activated B cells (NF-*κ*B), IFN-regulatory factor 3 (IRF3) resulting in transcription of proinflammatory genes and the subsequent inflammatory response [8,112].

TLR2 and TLR4 have polyvalent ligand binding activity and can be activated by exogeneous (e.g., lipopolysaccharide, LPS) and endogenous ligands comprising DAMPs released upon I/R. These DAMPs vary depending on type of injury and tissue involved. High-mobility group box-1 (HMGB-1), an intracellular protein involved in the organisation of DNA and the regulation of gene transcription, is one of the DAMPs linked to the pathogenesis of IRI [118,119,120]. From the nucleus, HMGB-1 can be released into the cytosol or extracellular space by passive leakage from injured cells or through active secretion by immune cells [121,122]. 

In IRI in the kidney, TLR4 plays an important role. Bergler et al. [123] showed that TLR4 is highly upregulated after renal IRI, and that high TLR4 expression is strongly correlated with graft dysfunction in an allogenic renal transplant model in rats. Furthermore, TLR4-deficient mice are protected against renal IRI and kidneys from donors with a TLR4-loss of function allele show less pro inflammatory cytokines in the kidney after transplantation and a higher percentage of immediate graft function [118,124]. Activation of TLR4 in renal IRI has various consequences on the graft. First of all it promotes the release of different proinflammatory mediators like IL-6, IL-1β and TNF-α, accompanied by an increased expression of macrophage inflammatory protein-2 (MIP-2) and monocyte chemo attractant protein-1 (MCP-1) involved in the recruitment of neutrophils and macrophages [124]. Second, TLR-4 activation leads to increased expression of adhesion molecules ICAM-1, vascular cell adhesion molecule 1 (VCAM-1) and E-selectin facilitating leukocyte migration and infiltration into the interstitial space. TLR-4 signalling seems mandatory for this increased expression. Chen et al. [125] showed that increased expression of adhesion molecules after renal IRI was absent in TLR4 knockout mice in vivo and the addition of HMGB-1 to isolated endothelial cells increased adhesion molecule expression on cells from wild-type but not from TLR4 knockout mice. Thirdly, activation of TLR4 on circulating immune cells of the innate immune system leads to activation of these cells. Neutrophils and macrophages are involved in an early stage after reperfusion. Neutrophils are regarded as the primary mediators of injury and their activation leads to ROS release, secretion of different proteases and renal tissue injury [126]. Upon activation, macrophages release proteolytic enzymes and proinflammatory cytokines like TNF-α, IL-1β and interferon-γ (IFN-γ) [127]. In TLR-4 knockout mice subjected to IRI, neutrophil and macrophage infiltration was reduced [124]. Finally, the TLR4-facilitated immune response is linked to renal fibrosis. The upregulation of TLR4 upon I/R induces a strong inflammatory response accompanied by tubular necrosis, loss of brush border, formation of casts and tubular dilatation [124]. Such a robust inflammation is known to potentiate interstitial fibrosis [128]. 

Proposed endogenous ligands for TLR-4 in renal IRI include HMGB-1, extracellular matrix (ECM) components like biglycan, heparin sulphate and soluble hyaluronan, and heat shock proteins (Hsps) [129,130,131,132,133,134]. Upon ligand binding, activation of TLR4 leads to downstream signalling via the MyD88-dependent and MyD88 independent pathway (Figure 6). The MyD88-dependent pathway in which MyD88 and TIRAP or MyD88 adapter-like (Mal) recruits and activates members of the IRAK family is considered to be the dominant pathway [124,135]. Wang et al. [136] demonstrated that MyD88- and TRIF-deficient mice showed a significant reduction in interstitial fibrosis reflected by α-SMA and collagen I and II accumulation Furthermore, Administration of the MyD88 specific inhibitor TJ-M2010-2, a small molecular compound, inhibiting the homodimerisation of MyD88, in a renal I/R model in mice has shown to prolong the survival rate, preserve renal function and attenuate the inflammatory responses and apoptosis in the kidney. In the long term, inhibition of the TLR/MyD88 signalling pathway with TJ-M2010-2 attenuated renal fibrosis via inhibition of TGF-β-induced epithelial to mesenchymal transition [137]. Liu et al. [138] showed that pre-treatment with the synthetic TLR4 inhibitor eritoran (Eisai co., Ltd, Tokyo, Japan) in an renal I/R rat model resulted in reduced expression of TNF-α, IL-1β and MCP-1, attenuated monocyte infiltration in the kidney and improved renal outcome Altogether in view of the pivotal role of TLR4 in renal IRI, inhibition of TLR4 or upstream or downstream mediators could be an interesting target in reducing IRI and optimising graft survival. 

Next to TLR4, TLR 2 is markedly upregulated upon ischemic injury in the kidney and its upregulation is associated with the initiation of an inflammatory response [139]. Kidneys of TLR2-/- mice subjected to I/R showed less tubular damage compared to TLR2+/+ mice. Reduced levels of MIP-2, MCP-1, and IL-6 and reduced levels of infiltrating leucocytes were seen [140]. The role of TLR2 in the development or progression of renal fibrosis, however, is less clear. Leemans et al. [139] showed that in a mouse model of obstructive nephropathy TLR2 does not play a significant role in renal progressive injury and fibrosis. In addition to this de Groot et al. [141] showed in human allograft biopsies that TLR2 expression 6, 12 and 24 months after transplantation is associated with superior graft outcome in the long run Currently, the humanized immune globuline (Ig) G4 (IgG4) monoclonal antibody against TLR2 OPN-305 (Tomaralimab, Opsona Therapeutics Ltd, Dublin, Ireland) has entered phase 2 trials (NCT01794663) with the aim to reduce delayed graft function in recipients of post-mortal donor kidneys. In the first part (A) of this study a single dose of 0.5 mg/kg administered 1h before reperfusion was associated with full inhibition of TLR2 and an 80% reduction of IL-6 [142]. Subsequently, this dose has been used in part B of the study, which has been completed but results have not been reported yet.

##### Complement System

The complement system is the second crucial player in the innate immune response in IRI. The system consists of soluble proteins, regulatory proteins and membrane-bound receptors and comprises three pathways. DAMPs released upon I/R are able to activate all three pathways via binding to C1q (classical pathway), C3 (alternative pathway) or PRRs of the lectin pathway (LP). 

Recently, the LP has been pointed out as the primary route of renal complement activation after I/R [143]. Activation of the LP can take place through various PRRs like collectins (manose binding lectin (MBL) and collectin-11) [144] and ficolins (ficolin 1-3) [145]. Upon binding of the collectin–mannan-binding lectin serine protease (MASP) complex to carbohydrate-bearing ligands (for instance, mannose or fructose expressed on stressed cells) the MASPs are activated to cleave complement component (C) 4 (C4) and C2. LP activation is critically dependant on the action of MASP-2 [146,147]. In an isograft transplantation model in wild-type and MASP-2-deficient mice, Asgari et al. [147] showed that renal function was preserved with MASP-2 deficiency After complex-ligand interaction, LP proceeds with cleavage of C4 and C2, mediated by MASP-2, leading to the synthesis of the classical pathway C3 convertase. Recently, a C4 independent bypass in the LP pathway was also demonstrated [122]. This could explain why C4-deficient mice are not protected against renal I/R and cellular mediated rejection [148,149]. One of the PRRs assigned an important role in the LP is collectin-11 (CL-11), a soluble C-type lectin containing a carbohydrate recognition domain and MASP binding domain [150]. In renal tissue, tubular cells are the main source of CL-11 and expression increases after IRI [151]. CL-11 has been appointed an important role in complement activation in the kidney. It has been shown that CL-11 engages l-fucose at sites of ischemic stress and inflammation initiating the LP [147]. In a renal I/R model, CL-11-deficient mice showed no post-ischemic and complement mediated injury supporting the importance of CL-11 in triggering renal complement activation.

All activating routes converge and lead to the formation of the C3 convertase (C4b2b, C3bBbP). C3 convertase cleaves and activates additional C3, creating C3a and C3b. C3b together with C4b2b forms the C5 convertase, which will cleave C5 into C5a and C5b. C5b together with C6–9 will then form the Membrane Attack Complex (MAC, C5b-9). The formed complement effectors will lead to opsonisation (C3b), chemotaxis of neutrophils and macrophages (C3a, C5a) [143]. The formed MAC inserted into the cellular membrane is associated with a proinflammatory response via noncanonical NF-ΚB signalling (Figure 7) [152,153].

Next to inducing inflammation and cell death, the complement system is able to modulate antigen presentation and T cell priming via C3a and C5a and is therefore playing a role in donor antigen sensitisation and rejection [154]. Antigen-presenting cells (APC) express C3 and C5 along with complement receptors C3aR and C5aR1. Upon complement activation in the extracellular space, C3a and C5a increase the presentation of alloantigens and expression of co-stimulatory molecules on the APC enhancing APC priming of T cells [143]. Furthermore, C3a and C5a promote T-cell differentiation of CD4+ and CD8+ T-cells. CD8+ cells mediate vascular and cellular T-cell mediated rejection. Upon activation, CD4+ T-cells can stimulate further CD8+ T-cell differentiation, they can proliferate and differentiate to memory and effector CD4+ cells which can activate macrophages, recruit leukocytes and stimulate inflammation and finally CD4+ cells stimulate B-cell differentiation and in the end antibody production [143]. The B-cells response can also be enhanced in a direct manner via C3b and C3d on the APC and the complement receptor 2 (CR2) on the B-cell. Activation of the B-cell by binding to the donor alloantigen induces class switching of the donor specific antibody from IgM to IgG. Subsequently, ABMR occurs when IgG donor specific antibodies (DSA) recognizes antigens in the kidney graft and engage with C1q, C1r and C1s to activate the classical pathway [143]. Under normal physiological circumstances, formation of the complement effectors is controlled by proteins (soluble or surface bound) that mediate break down of the C3 and C5 convertases. After I/R this balance shifts to uncontrolled complement activation predisposing the graft to complement mediated injury and rejection [155]. 

Many interventions on the level of C3, C5, and regulatory proteins in I/R injury and especially kidney transplantation have been evaluated in pre- and clinical studies [156]. Eculizumab (Soliris®, Alexion Pharmaceuticals, New Haven, CT, USA) is to date the best studied complement inhibitor in kidney transplantation. Therapeutic inhibition of C5 with the use of eculizumab, an anti-human C5 micro antibody, showed potential in the prevention and/or treatment in AMBR [157,158,159] and has been investigated as such in several phase 2/3 clinical trials (NCT01567085, NCT01106027, NCT01399593). All studies report a safety profile of the drug that is consistent with that reported for eculizumab’s approved indications like atypical haemolytic uremic syndrome. Results of these trials suggest a potential role of eculizumab in the prevention and treatment of ABMR in patients with DSA [160,161]. Next to ABMR, eculizumab has been investigated for the prevention of DGF (NCT01919346, NCT02145182). Again, the safety profile was good but pre-treatment with eculizumab had no effect on the incidence of DGF. Groups in these studies, however, were rather small [162]. Another anti-C5 antibody Tesidolumab (LFG-316, MorphoSys, Novartis) has currently entered phase 1 studies (NCT02878616). 

In addition to targeting terminal complement pathways, therapeutics targeting complement initiation (C1) and amplification (C3, convertases) have been developed. C1 esterase inhibitors (C1-INH) should not be considered complement-specific inhibitors, since these broad protease inhibitors and their functions extend beyond the classical pathway and even beyond the complement system [163]. The C1INH Cinryze® (Shire US Inc., Lexington, MA, USA) is recently being evaluated for treatment of ABMR (NCT02547220). The study was terminated May 2019 following a pre-scheduled interim analysis, it was determined that the study met the pre-specified criteria for futility. Cinryze® is still listed to be tested as a pre-treatment to reduce IRI and DGF (NCT02435732). Another C1INH, Berinert® (CSL Behring, King of Prussia, PA, USA), has been evaluated in a phase 1/2, double-blind, placebo-controlled study assessing its safety and efficacy for prevention of delayed graft function in recipients of deceased donor kidneys [164]. Although the primary outcome measure (DGF) was not met, treatment with Berinert® was associated with significantly fewer dialysis sessions 2 to 4 weeks post-transplantation. In addition, a better renal function was seen at 1 year compared with the placebo treated group. No significant adverse events were noted in this study [164]. Finally, Mirococept (APT070) a membrane-localising C3 convertase inhibitor is currently being evaluated in a double-blind randomised controlled investigation its efficacy for preventing IRI deceased donor kidneys (EMPIRIKAL-trial, ISRCTN49958194) [165].

##### Translation to the Adaptive Immune System

The link between the innate and adaptive immune response is made by dendritic cells (DCs, Figure 8). DCs are APCs and play an essential role in the pathogenesis of IRI. Immature DCs can be activated by DAMPs via TLRs and the complement system. After maturation, they are able to activate the adaptive immune system in a direct manner by antigen presentation to B- and T-cells or indirectly via cytokine signalling [8,166]. This process can already start in the donor in which in case of a DBD donor, DCs are activated by oxidative stress or C5a and present donor antigens to T-cells of the recipient [167]. Furthermore, it is thought that DCs (subtype CDC11c+ and F4/80+) play an important role in the early pathophysiology of IRI by secretion of TNF-α, Chemokine (C-C motif) ligand 5 (CCL5), IL-6 and MCP-1within the first 24h after IRI [168]. Further, at a later stage, DCs contribute to allograft dysfunction. Batal et al. [169] looked at kidney transplant biopsies performed > 15 days after transplantation and found that a high DC density was independently associated with poor graft survival. Additionally, they found that high DC density was correlated with an increased T-cell proliferation and poor patient outcome in patients with high total inflammation scores of biopsies, including inflammation in areas of tubular atrophy. In these patients, DC density could predict allograft loss. When looking at the origin of the DCs they showed that initially donor DC predominated but found that in late biopsies the majority of DCs were of recipient origin. These data suggest a potential rationale to target DCs influx in the kidney to improve long-term allograft survival.

#### 3.3.2. Adaptive Immune Response

In contrast to the non-specific nature of the innate immune response, the role of the adaptive immune system is to recognize alloantigens and to react with an alloantigen-specific response, simultaneously generating immunological memory. Involved cells are B- and T-cells. 

##### T-Cells

Activation of T-cells occurs through binding of the T-cell receptor (TCR) on the surface of the T-cell, to the major histocompatibility complex (MHC, in case of humans the human leucocyte antigen (HLA) system) on the APC. This can be in a direct way when the TCR binds to unprocessed allogenic MHC on the APC of the donor or in an indirect manner when MHC proteins of the donor have been taken up by APC of the recipient, processed and presented by the MHC of the recipient [170]. In case of IRI, CD4+ T helper (Th) cells as well as CD8+ cytotoxic T-cells are found in the kidney and are important mediators of IRI [171,172,173,174]. T-cell-deficient mice showed attenuated renal IRI and adoptive T-cell transfer experiments in athymic mice resulted in acute kidney injury (AKI) [175,176,177]. 

The TCR on CD4+ T-cells can only bind to MHC class 2 molecules (HLA DP, DQ, DR). Upon activation, these CD4+ T-cells become cytokine producing effector cells harming the graft through cytokine mediated inflammation [170]. The effector CD4+ Th cells can differentiate into three major subtypes Type 1 (Th1), Type 2 (Th2) and Th17 cells depending on the cytokines they produce and the transcription factors they express. This differentiation process, referred to as polarisation, starts with induction in lymphoid tissue. Cytokines produced by APCs (DCs and macrophages), NK cells, basophils and mast cells act on T-cells stimulated by the antigen and co-stimulators. This induces transcription of cytokine genes characteristic for the particular subset. Upon continued activation, genetic modifications occur, keeping the characteristic cytokine genes in a transcriptionally active state (commitment). The cytokines produced by the subset promote development of this subset and inhibit differentiation toward other subsets (amplification) [170]. The main effector cytokine of Th1 cells is IFN-γ and the key Th1 transcription factors are signal transducer and activator of transcription (STAT) 4 (STAT-4) and the T-box transcription factor T-bet. Main effector cells are macrophages, B-cells, CD8+ T-cells and CD4+ T-cells (amplification). IFN-γ secreted by Th1 cells will activate macrophages leading to secretion of inflammatory cytokines (TNF, IL-1 and IL-2), an increased production of toxic substances like ROS, NO and lysosomal enzymes and finally stimulation of expression of costimulatory molecules enhancing the efficiency of the macrophage as APC [170]. The main effector cytokines of Th2 are IL-4, IL-5 and IL-13 and key transcription factors are GATA binding protein 3 (GATA-3) and STAT-6. IL-4 act on B-cells to stimulate production of IgE antibodies which can lead to mast cell degranulation upon binding of IgE with mast cells. IL-5 activates eosinophils, inducing defence against helminthic infections. IL-4 and IL-13 are involved in alternative macrophage activation promoting development of M2 macrophages which have anti-inflammatory effects and may promote tissue repair and fibrosis [170]. Signature cytokines of Th17 are IL-17 and IL-22. Differentiation into this subtype is mediated by IL-6 and TGF-β leading to activation of transcription factors STAT-3 and retinoic acid-related orphan receptor γ*t* (RORγ*t*) respectively. IL-17 act on leukocytes and tissue cells and stimulates production of several chemokines and cytokines (TNF-α, IL-1β, IL-6) that recruit neutrophils and to a lesser extend monocytes to generate an inflammatory response. IL-22 produced in epithelial cells is primarily involved in maintaining the barrier function of epithelia [170]. Th17 T-cell most likely play a significant role in IRI-induced inflammation. STAT-3 KO mice are protected from renal IRI via downregulation of Th17 activity [178]. The differentiated T-cells can convert from one subtype to another by changes in activation circumstances [179]. It is suggested that Th1/Th2 ratio plays an important role in the pathogenesis of IRI [180,181]. Yokota et al. [181] demonstrated that STAT-6-deficient mice with a defective Th2 phenotype have enhanced renal I/R injury whereas STAT-4-deficient mice have mild improved function In addition, Loverre et al. [182] showed that kidney transplant recipients experiencing DGF predominantly expressed Th1 phenotype within the graft In literature both Th1 and Th17 cells are associated with T-cell mediated rejection [183,184,185,186,187,188]. 

The TCR on CD8+ T-cells can only bind to MHC class 1 molecules (HLA A, B, C) presented on APCs. Upon activation in lymphoid tissue, they differentiate into cytotoxic T-cells (CTLs) or memory cells. This differentiation is facilitated by CD4+ Th1 cells by secreting cytokines that act directly on the CD8+ cells [170]. The main cytokines involved are IL-2 (proliferation, differentiation CTL/memory cell), IL-12/IFN (differentiation CTL), IL-15 (memory cell survival), IL-21 (memory cell induction). The CTLs are able to kill cells which present the allogenic class 1 MHC of the donor in the graft. This through binding on the target cell and release of granule content into the immune synapse. These granules contain perforin and granzymes. Perforin induces the uptake of granzymes into the target cell. These granzymes are capable of activating caspases and inducing apoptosis. The killing of the target cell can also be Fas/Fas-L mediated in which the CTL expose the Fas ligand on the membrane which will bind to the Fas receptor on the target cell inducing apoptosis. Only CTLs that are activated in the direct way (by donor MHC on donor APC) are able to kill graft cells [170]. Like CD4+ Th cells, CTL secrete inflammatory cytokines, (predominantly IFN-γ) that attribute to inflammation and injury of the graft. The role of CD8+ cells in early phase of renal IRI is unclear, in a mouse model CD4+ deficient mouse was protected from IRI but CD8+ deficient mouse was not [176]. 

Ko et al. [189] showed that already 6 h after renal IRI, transcriptional activity occurs in T-cells and that these gene expression changes persist up to 4 weeks after the event. Genes involved in immune cell trafficking and cellular movement were most upregulated in the early phase (6 h, 3 days). On day 10 this was shifted to genes related to cellular development products involved in immune responses and on day 28 to genes involved in cellular and humoral immune response involved in antigen presentation. In addition, they found that the CC motif chemokine receptor 5 (CCR5) was one of the most upregulated genes at all time points, which was confirmed at a protein level. Subsequently, the addition of CCR5 antibody attenuated IRI and led to decreased T-cell activation [189]. 

##### B-Cells

Next to alloreactive CD4+ and CD8+ T-cells, antibodies (immune globulins, Ig) against the graft contribute to rejection. Most of these Igs are produced by Th dependant alloreactive B-cells. The naive B-cell recognizes allogenic MHC-molecules, processes these MHC-molecules and presents them to Th cells that were activated previously by the same alloantigen presented by APCs. The produced Igs (IgM/IgG) are then able to induce complement activation, and activation of neutrophils, NK cells and macrophages. The T-cells are responsible for T-cell mediated rejection and B-cells together with complement activation for ABMR [170].

##### Regulatory T-Cells

The T-cells which most likely play a protective role in renal IRI are regulatory T-cells (Tregs), a subset of CD4+ T-cells whose function is to supress the innate as well as the adaptive immune response and maintain self-tolerance. Tregs can be discriminated from other T-cells by expression of FoxP3 amongst other proteins like CD25. FoxP3 is probably the most important transcription factor for Treg differentiation. The mechanism of action of Tregs is production of immune suppressive cytokines IL-10 and TGF-β, reduction of APC is to stimulate T-cells (possibly by binding to B7 proteins on the APC) and finally consumption of IL-2, an important growth factor for other T-cells [170]. TGF-β inhibits various immune cells amongst which: proliferation and effector functions of T-cells, macrophages, neutrophils and endothelial cells. It regulates differentiation of FoxP3+ Tregs and promotes polarisation towards Th17 cells. Furthermore, TGF-β promotes tissue repair by the ability to stimulate collagen synthesis and matrix modifying enzyme by macrophages and fibroblasts. IL-10 inhibits the production of IL-12 by activated macrophages and DCs, therefore inhibiting these cells and their IFN-γ production. It also inhibits T-cell activation by inhibiting the expression of co-stimulators and MHC-II molecules on DCs and macrophages [170]. 

Tregs play a potentially promising role in the reduction of IRI and graft tolerance [190,191,192,193]. Currently, several clinical trials are running evaluating the safety and effeciacy of FoxP3 cellular therapy in kidney transplantation (NCT02091232, NCT03284242, NCT01446484) [194,195]. However, all that glitters is not gold, since recent studies have shown that human FoxP3+ T-cells show great variations in gene expression phenotype and function [196,197,198,199]. Furthermore, recently a subset of FoxP3+ Tregs mimicking Th cells was discovered that secreted pro-inflammatory cytokines [200]. Also, the effect of different immune suppressive agents on the Treg phenotype needs to be elucidated, since these drugs might influence Treg phenotype [200,201]. Altogether, more insight in function and biology is needed before this therapy finds its way to clinical settings.

### 3.4. Transcriptional Reprogramming

Finally, cells can protect themselves from hypoxia and ischemia and maintain homeostasis via an evolutionary conserved mechanism with the use of oxygen sensors and activation of specific transcription factors. These so called hypoxic inducible factors (HIFs) regulate various genes involved in the metabolic cell cycle, angiogenesis, erythropoiesis, energy conservation and cell survival and are therefore able to induce a protective cell response to hypoxia [202]. 

HIFs are heterodimeric transcription factors consisting of an α and β subunit. There are two types of α subunits, HIF-1α and HIF-2α, which have common, but also subunit-specific target genes. In the kidney, HIF-1α is predominantly localized in tubular and glomerular cells, whereas HIF- 2α can be found in glomerular cells, peritubular endothelial cells and fibroblasts [203,204,205]. In aerobic circumstances, HIFs are inactive. Oxygen-sensing prolylhydroxylase (PHD) hydroxylates the amino acid proline on the HIF-1α/HIF-2α subunit. This induces a conformational change enabling von Hippel–Lindau tumour suppressor protein (pVHL) to bind with the α-subunit, leading to degradation of the HIF-α subunit. Ischemia/hypoxia will lead to inhibition of the oxygen-dependent PHD, which enables nuclear translocation of the α subunit, binding of the α and β subunit and formation of HIF. In the nucleus HIF binds with the hypoxia response promotor element (HRE) leading to the transcription of various genes like glycolysis enzymes Glut-1 and aldolase (enabling ATP production under hypoxic circumstances), NF-*κ*B, TLRs, adenosine receptors, vascular endothelial growth factor (VGEF), CD73 and erythropoietin. Activation of HIF can also occur in normoxemic circumstances, for instance, by ROS, LPS, various cytokines and TCR-CD28 stimulation. Transcriptional reprogramming is a consequence of I/R that should be considered a protective mechanism (Figure 9) [206]. 

Conde et al. [207] showed in various models and human post-transplantation biopsies that HIF-1α is induced in a biphasic manner namely during the hypoxic as well as the reperfusion phase. They pointed out the PI3K/Akt mTOR pathway to be responsible for this HIF-1α accumulation during the normoxemic reperfusion phase. In their study, this second increase (e.g., during reperfusion) seemed crucial for tubular cell survival and recovery. During the hypoxic phase, an increase in HIF-1 resulted predominantly in the upregulation of PHD3 and VGEF mRNA, which remained elevated during oxygenation. EPO mRNA was upregulated upon reperfusion. EPO and VGEF have been suggested to be involved in proximal tubular regeneration [208,209,210]. Their human post-transplantation biopsies revealed HIF-1α expression in proximal tubular cells without ischemic damage or features of regeneration suggesting a protective role for HIF-1α during I/R [207]. Oda et al. [211] had similar findings in patients receiving a DBD/DCD donor kidney. Their analysis of 46 post-transplant biopsies, gained 1h after reperfusion, showed that expression levels of PI3K, Akt, mTOR and HIF-1α were significantly higher in patients without DGF compared to patients experiencing DGF (76% of the patients). The expression levels of HIF-1α and donor type (DCD) were independently associated with DGF HIF-2α expression in renal endothelial cells is suggested in several studies to be protective against renal IRI via protection and preservation of the vasculature endothelium by upregulation of angiogenic factors like VGEF and their receptors Tie2 and VGEFreceptor-2 (FLK-1) [212,213,214,215]. Increased production of HIF in myeloid and lymphoid cells influences the innate and adaptive immune response. T-cell activation and proliferation is reduced under hypoxic conditions [216]. A study of Zhang et al. [217] revealed a hypoxia/HIF-2α/adenosine2A receptor axis to be responsible in reduction of NK T-cells activation and renal IRI upon I/R. HIF-1α induces a shift from Th1 to Th2 cells (decrease Th1/Th2 ratio) accompanied by a decrease in excretion of inflammatory cytokines. Furthermore, HIF-1α promotes transcription of FoxP3 and therefore generation activation of Tregs. 

Various PHD inhibitors have been developed and tested in animal I/R models. In a rat model, Wang et al. [218] showed that use of the PHD-1 inhibitor acetate prior to the ischemic event was able to stabilize HIF in a dose-dependent manner and was associated with improved renal outcome. In addition, in an allogenic renal transplant model in rats, the use of the PHD inhibitor FD-4497 pre-donation was associated with increased HIF expression and improved graft outcome and reduced mortality of recipients [219]. Hence, activation and/or upregulation of HIF could be an interesting approach to reduce renal IRI and improve renal transplant outcome. Several PHD inhibitors are currently being tested in clinical trials in order to treat anaemia in patients with chronic kidney disease but have not been tested in the field of transplantation yet.

## 4. Summary

The past decade’s research in kidney transplant recipients has focussed on post-transplant patient management, with a predominant emphasis on immunosuppression. However, the biggest ‘hit’ to the donor organ is encountered during the process of donation and reperfusion at time of transplantation, i.e., ischemia and reperfusion injury. An important initiating step in IRI is the uncontrolled ROS formation during reperfusion and dysfunction of the mitochondrial machinery leading to the opening of mPTP and the release of DAMPs in the intra- and extracellular space. From here, several injury cascades are activated, including activation of cell death programs like apoptosis and (regulated) necrosis, endothelial dysfunction implicating increased vasoconstriction upon reperfusion, loss of specific phenotype of endothelial cells and transmigration of leucocytes into the interstitial space. Activation of the innate and subsequently the adaptive immune system will take place through binding of DAMPs to the toll-like receptors and activation of the complement system, leading to further injury of the graft, increased immunogenicity favouring T-cell and antibody mediated rejection and the initiation of fibrosis associated with chronic graft dysfunction. Currently, several novel agents targeting various pathways are tested and, although most are still in the preclinical phase, some have already entered clinical trials. Intervention early in this cascade of events (e.g., on a mitochondrial level), seems very attractive, since mitochondrial dysfunction plays a pivotal role in the initiation of IRI. Due to the complexity of the pathophysiological mechanisms, however, it may be predicted that a multiple treatment strategy using a combination of agents given at various time points during the donation, preservation and transplantation process will most likely be the best strategy to reduce IRI. 

## Figures and Tables

**Figure 1 jcm-09-00253-f001:**
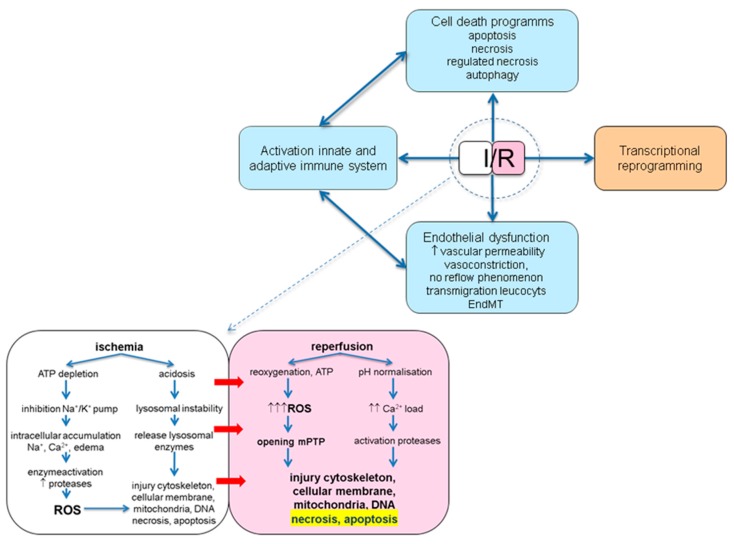
Schematic overview of the pathophysiological consequences of ischemia and reperfusion. I/R: ischemia/reperfusion; ATP: adenosine triphosphate; EndMT: endothelial to mesenchymal transition; ROS: reactive oxygen species; mPTP: mitochondrial permeability transition pore.

**Figure 2 jcm-09-00253-f002:**
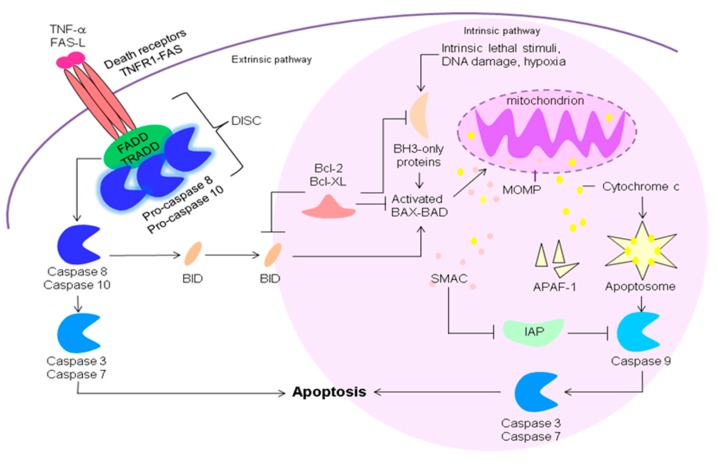
Extrinsic and intrinsic apoptotic pathway. The intrinsic pathway is mediated by intracellular signals of cell stress leading to an increase in the BH3 only proteins (members of the B-cell lymphoma 2 (Bcl-2) family) resulting in an inhibition of the protectors and activation of the effectors. The effectors Bax and Bad increase the permeability of the mitochondrial membrane (MOMP: mitochondrial outer membrane permeabilisation) resulting in leakage of apoptotic proteins. One of these proteins, known as second mitochondria-derived activator of caspases (SMAC), binds to proteins that inhibit apoptosis (IAPs, by suppression of the caspase proteins) causing an inactivation of these IAPs. Another protein released from the mitochondria is cytochrome c, which binds to Apoptotic protease activating factor-1 (Apaf-1) and ATP. This complex binds to procaspase-9 creating a complex, the apoptosome. The apoptosome cleaves procaspase-9 to its active form of caspase-9, which in turn is able to activate the effector caspase-3. The extrinsic pathway is mediated through receptors of the tumor necrosis factor (TNF) receptor (TNFR) family either via the TNF path or the Fas (first apoptosis signal) path. In the TNF path binding of TNF-α to a trimeric complex of TNFR1 molecules induces activation of the intracellular death domain and the formation of the receptor-bound complex 1 made up of TNF receptor-associated death domain (TRADD), receptor-interacting protein kinase 1 (RIPK1), two ubiquitin ligases (TNFR-associated factor (TRAF)-2 and cellular inhibitors of apoptosis (clAP)1/2) and the linear ubiquitin assembly complex (LUBAC). This complex 1 can lead to a pro-survival pathway or to apoptosis. In case of apoptosis the TRADD dependant complex IIa (consisting of TRADD, Fas-associated death domain protein (FADD) and caspase-8) or the RIPK-1 dependant complex IIb also known as the ripoptosome (consisting FADD, RIPK1, RIPK3 and caspase-8) is formed. In the Fas path, presence of the Fas ligand (FasL, expressed on cytotoxic T lymphocytes) causes three Fas receptors (CD95) to trimirize. This clustering and binding to the FasL initiates binding of FADD. Three procaspase-8 or -10 molecules can then interact with the complex by their own death effector domains. The complex formed is the death-inducing signalling complex (DISC) which cleaves and activates procaspase-8 and 10. Activation of the initiator caspase-8 by both paths directly activates other members of the caspase signalling cascade such as the effector caspase-3 but also can lead to an increase in BH3-only proteins (Bim, Bid) and trigger the intrinsic pathway).

**Figure 3 jcm-09-00253-f003:**
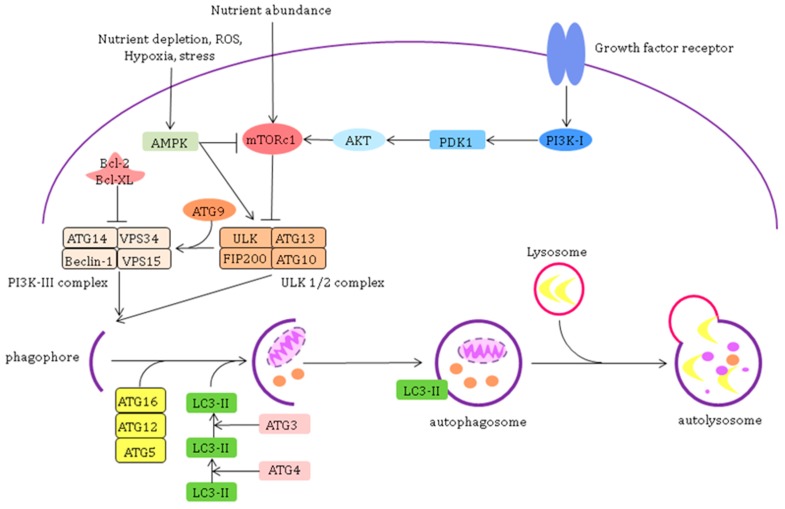
Pathways of macro-autophagy. Initiation of autophagy is regulated by mTORC1 (mammalian target of rapamycin complex 1) and AMPK (AMP-activated kinase). Together, they regulate the activity of the ULK1/2 complex consisting of ULK1/2 (Unc-51 like autophagy activating kinase), FIP200 (FAK family kinase interacting protein of 200 kDa) and the autophagy related proteins (ATG) ATG13 and ATG10. Activation of mTOR leads to the phosphorylation of this complex and inhibition of autophagy (for instance, through the phosphatidylinositol 3-kinase (PI3K)/ Protein kinase B (AKT) or the mitogen-activated protein kinase (MAPK)/ extracellular signal–regulated kinase (Erk) 1/2 signalling pathway) whereas activation of AMPK activates autophagy. AMPK, activated upon intracellular AMP increase, is able to activate autophagy by inhibition of the mTORC1 through dissociation of mTORC1 from ULK1/2 allowing ULK1/2 to be activated. AMPK, is also able to initiate autophagy in a direct way by phosphorylation of ULK1/2 forming the ULK1/2-complex.Another complex involved in the initiation is the autophagy inducible beclin-1 complex (or class III PI3K complex) which consists of Vps34 (phosphatidylinositol 3-kinase), beclin-1 (a BH3 only domain protein member of the Bcl-2 family), vps15 and ATG14. This complex is activated by the ULK-1 complex and inhibited by Bcl-2 and Bcl-XL. The ULK1/2 and class III PI3K complexes join to form the phagopore and eventually the autophagosme. This process is mediated by the ATG5-ATG12-ATG16 complex and the formation of phosphatidylethanolamine-conjugated Light Chain (LC) 3 (LC3-II) facilitating elongation of the bilipid membrane to form a closed autophagosme. The autophagosome fuses with a lysosome and the content of the autolysosome is degenerated and the components are released to be reused to synthesise new proteins or to function as an energy source for the cell. PDK-1: pyruvate dehydrogenase kinase-1.

**Figure 4 jcm-09-00253-f004:**
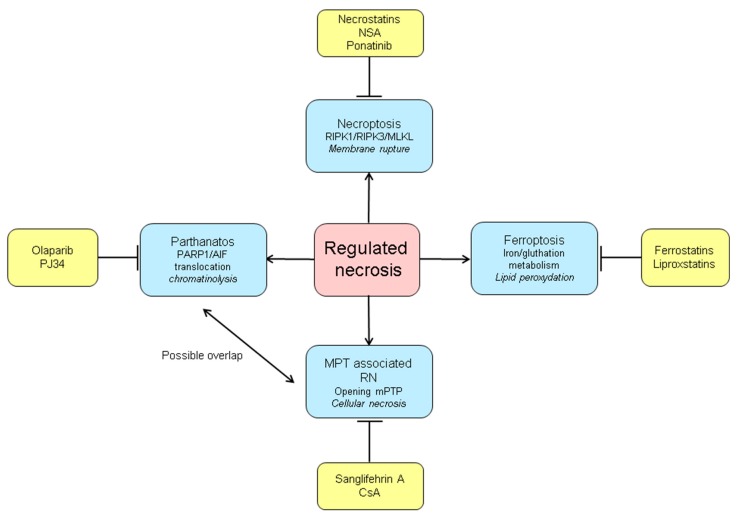
Programs of regulated necrosis and their inhibitors. RIPK1: receptor-interacting protein kinase 1; RIPK3: receptor-interacting protein kinase 3; MLKL: Mixed Kinase Domain-Like protein; MPT: mitochondrial permeability transition; mPTP: mitochondrial permeability transition pore; RN: regulated necrosis; CsA: cyclosporin A; PARP1: poly (ADP-ribose) polymerase-1; AIF: apoptosis-inducing factor.

**Figure 5 jcm-09-00253-f005:**
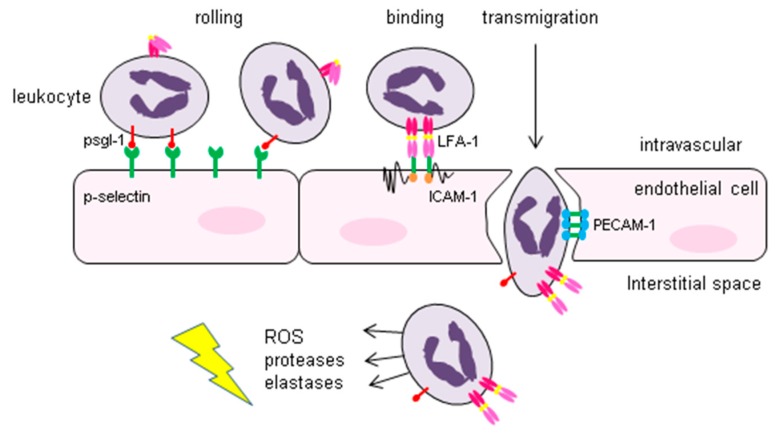
Interaction of leukocytes and endothelial cells in the process of transmigration of leukocytes. The increased expression of P-selectin on the endothelial cells upon I/R facilitates interaction with P-selectin glycoprotein 1 (PSGL-) expressed on the leukocytes. This results in rolling of the leukocytes on the endothelium. Subsequently, firm adherence of the leucocytes to the endothelium is achieved by interaction of lymphocyte function-associated antigen 1(LFA-1) and macrophage-1 antigen (MAC-1 or complement receptor 3, CR3) on the leukocyte and the intracellular adhesion molecule 1 (ICAM-1) on the endothelial cells. Finally, platelet endothelial cell adhesion molecule 1 (PECAM-1) facilitates transmigration of the leukocytes into the interstitial space. Once activated, these leukocytes will release several toxic substances like ROS, proteases, elastases and different cytokines in the interstitial compartment resulting in further injury like increased vascular permeability, oedema, thrombosis and parenchymal cell death.

**Figure 6 jcm-09-00253-f006:**
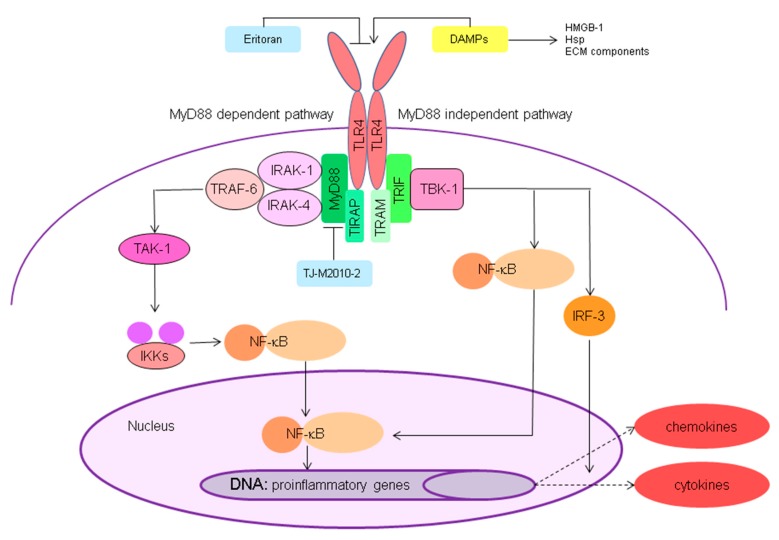
Toll-like receptor 4 signalling. Activation of toll-like receptor 4 (TLR4) by danger associated molecular patterns (DAMPs), like high mobility group box-1 (HMGB-1), heat shock proteins (hsp) and extracellular matrix (ECM) components, leads to downstream signalling via the MyD88 (Myeloid differentiation primary-response protein 88) dependent and MyD88 independent pathway. The MyD88-dependent pathway in which MyD88 and TIRAP (toll-interleukin 1 receptor (TIR) domain containing adaptor protein) or MyD88 adapter-like (Mal) recruits and activates members of the IL-1 receptor-associated kinase (IRAK) family is considered to be the dominant pathway. IRAK activation leads to recruitment of TRAF6 (TNF receptor-associated factor 6) and subsequently activation of transforming growth factor beta-activated kinase 1 (TAK1). Activation of TAK1 then leads to the activation of inhibitor of nuclear factor-κB kinase (IKK), which results in the release of nuclear factor kappa-light-chain-enhancer of activated B cells (NF-κB) from its inhibitor, promoting translocation to the nucleus. The MyD88 independent pathway is mediated by the adapter molecules TIR-domain-containing adapter-inducing interferon-β (TRIF)/TRIF-related adaptor molecule (TRAM) and downstream signalling leads to activation of 2 inhibitor of nuclear factor-κB kinase (IKK) homologs IKKε and TANK-binding Kinase-1 (TBK1), which possibly form a complex together and activate transcription factors NF-κB and IFN-regulatory factor 3 (IRF3). From here, proinflammatory gene transcription is initiated. TLR4 signalling is inhibited by Eritoran and TJ-M2010-2.

**Figure 7 jcm-09-00253-f007:**
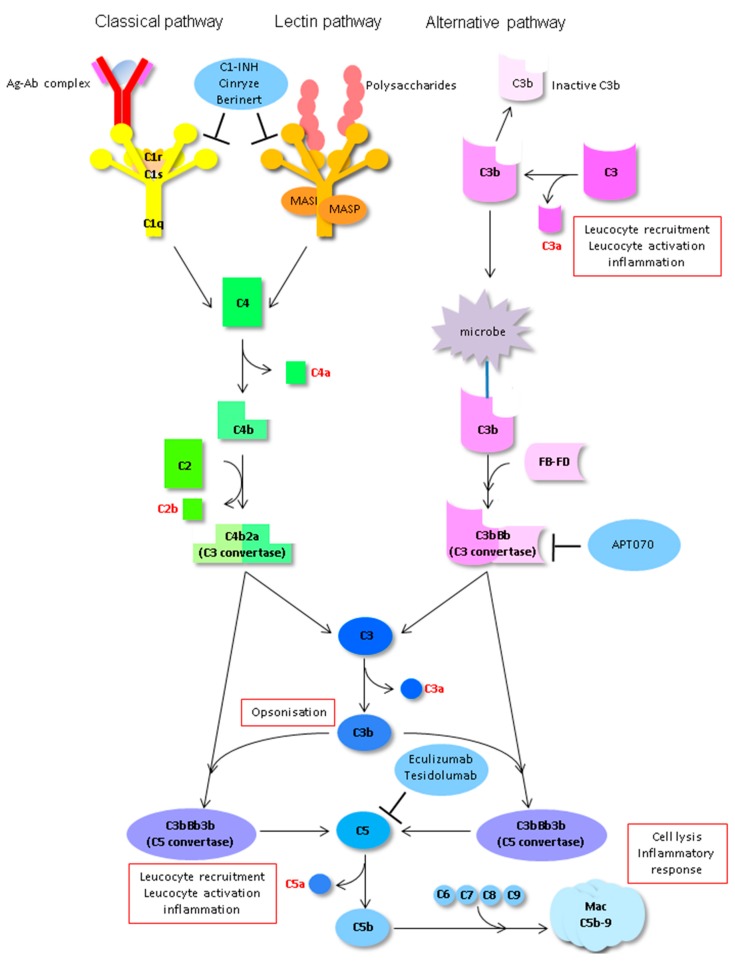
Routes of the complement system with its inhibitors currently studied in kidney transplantation. Damps released upon I/R are able to activate all three pathways via binding to C1q (classical pathway), C3 (alternative pathway) or pattern recognition receptors (PRRs) of the lectin path. All activating routes converge and lead to the formation of the complement component (C) 3 (C3) convertase (C4b2b, C3bBbP). C3 convertase cleaves and activates additional C3, creating C3a and C3b. C3b together with C4b2b forms the C5 convertase, which will cleave C5 into C5a and C5b. C5b together with C6–9 will then form the Membrane Attack Complex (MAC, C5b-9). The formed complement effectors will lead to opsonisation (C3b), chemotaxis of neutrophils and macrophages (C3a, C5a). The formed MAC inserted into the cellular membrane is associated with a proinflammatory response via noncanonical NF-ΚB signalling. C1-inhibitors (C1-INH), Cinryze® and Berinert® target complement initiation and APT070 complement amplification. Eculizumab and Tesidolumab inhibit complement activation at the level of C5.

**Figure 8 jcm-09-00253-f008:**
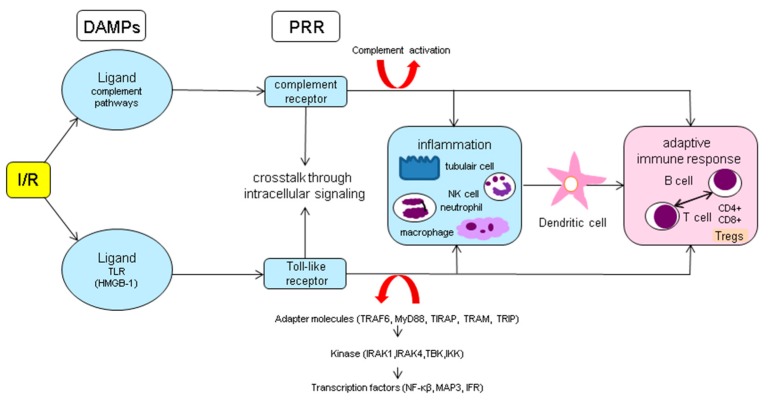
Interaction of the innate and adaptive immune system in the pathophysiology of ischemia and reperfusion injury. DAMPs released upon I/R are able to activate the innate immune system by binding to PRRs like complement receptors and TLRs. Activation of these receptors will lead to production of pro-inflammatory cytokines and chemokines and chemotaxis, opsonisation and activation of leucocytes like macrophages, neutrophils and natural killer (NK) cells. Additionally, immature dendritic cells can be activated, which, after maturation, are able to activate the adaptive immune system in a direct manner by antigen presentation to B- and T-cells or indirectly via cytokine signalling. Treg: regulatory T-cell.

**Figure 9 jcm-09-00253-f009:**
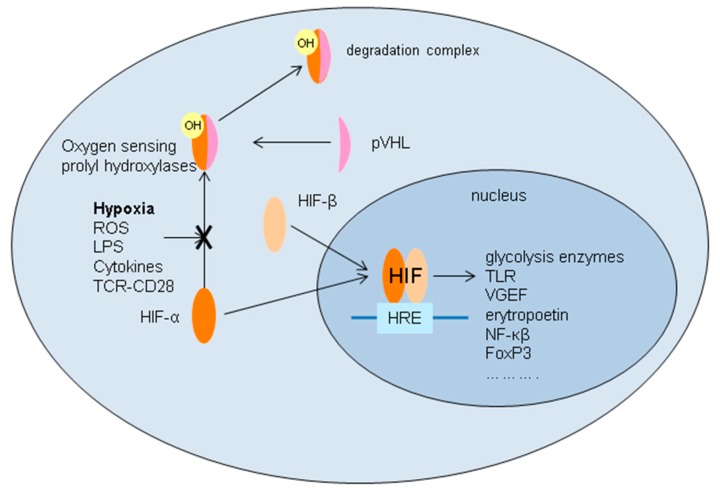
Intracellular stabilisation and activation of hypoxic inducible factor. Under normoxemic conditions, proline on the hypoxic inducible factor (HIF) α (HIFα) subunit is rapidly hydroxylated by oxygen-sensing prolyl hydroxylase (PHD). This induces a conformational change enabling von Hippel–Lindau tumour suppressor protein (pVHL) to bind with the α-subunit, leading to degradation of the HIF-α subunit. Ischemia (or other signals like lipopolysaccharide (LPS), various cytokines, etc.) will lead to inhibition of the oxygen-dependent PHD, enabling nuclear translocation of the α subunit, binding of the α and β subunit and formation of HIF. In the nucleus, HIF binds with the hypoxia response promotor element (HRE) leading to the transcription of various genes. VGEF: vascular endothelial growth factor.

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
