# Peer review of "Ischemia and Reperfusion Injury in Kidney Transplantation: Relevant Mechanisms in Injury and Repair"

_jcm, 2020, doi:10.3390/jcm9010253_

Round 1
Reviewer 1 Report
This is a comprehensive review on ischemia and reperfusion in kidney transplantation. The review details the pathophysiological processes involved in renal ischemic and reperfusion injury. It also highlights new therapeutic interventions used to improve kidney graft outcome
This is a well written and extremely comprehensive review on ischemia and reperfusion injury in kidney transplantation. The review is structured in a format detailing the pathophysiological and molecular processes. Each section provides a broad overview followed by a detailed description of the pathways involved, recent research findings and therapeutic agents currently being trialled.
Whilst the topics are informative, the level of detail used in each of the sections makes the review difficult to read in places. In each section a brief summary of the mechanisms could be followed by more highlight on the more recent findings. The figures could be used to provide detail and description of the molecular mechanisms and pathways without adding to the text.
Specific comments
The legend of the figures should summarise the mechanistic actions and effects as described in figure 6.
A figure describing the complement system and the adaptive immune response would be helpful.
Author Response
Dear editors and reviewers,
Thank you very much for your thorough review of our manuscript. We have adapted most of your suggestions. To our opinion your suggestions significantly contributes to the quality of the manuscript. All the changes are marked in red.
Response to reviewer 1.
Whilst the topics are informative, the level of detail used in each of the sections makes the review difficult to read in places. In each section a brief summary of the mechanisms could be followed by more highlight on the more recent findings. The figures could be used to provide detail and description of the molecular mechanisms and pathways without adding to the text.We would like to thank the reviewer for this point. We have redirected some of the text in the paragraphs to the figures and provided the sections with a brief summary when possible The legend of the figures should summarise the mechanistic actions and effects as described in figure 6.
A figure describing the complement system and the adaptive immune response would be helpful. We have added a figure describing the complement system
Reviewer 2 Report
Very well written. Good details on molecular pathways with diagrams.
It would be useful to have limited mention on the current clinical and pharmacological intervention to address IRI and their level of action on the pathways.
Author Response
Dear editors and reviewers,
Thank you very much for your thorough review of our manuscript. We have adapted most of your suggestions. To our opinion your suggestions significantly contributes to the quality of the manuscript. All the changes are marked in red.
Response to reviewer 2.
It would be useful to have limited mention on the current clinical and pharmacological intervention to address IRI and their level of action on the pathways. We understand the reviewers point of view but due to the length of the manuscript we decided to only discuss potential therapeutic strategies specifically targeting the molecular pathways of IRI. We added a sentence on this in the introduction. If we would also describe other strategies like ischemic/pharmacological conditioning, machine perfusion, donor management etc the manuscript would become enormous….